# Echoes Beyond Points: Unleashing the Power of Raw Radar Data in Multi-modality Fusion

**Yang Liu**
CASIA

**Feng Wang**
TuSimple

**Naiyan Wang**
TuSimple

**Zhaoxiang Zhang**
MAIS of CASIA & UCAS,
HKISI_CAS

{liuyang2022, zhaoxiang.zhang}@ia.ac.cn   {feng.wff, winsty}@gmail.com

## Abstract

Radar is ubiquitous in autonomous driving systems due to its low cost and good adaptability to bad weather. Nevertheless, the radar detection performance is usually inferior because its point cloud is sparse and not accurate due to the poor azimuth and elevation resolution. Moreover, point cloud generation algorithms already drop weak signals to reduce the false targets which may be suboptimal for the use of deep fusion. In this paper, we propose a novel method named EchoFusion to skip the existing radar signal processing pipeline and then incorporate the radar raw data with other sensors. Specifically, we first generate the Bird's Eye View (BEV) queries and then take corresponding spectrum features from radar to fuse with other sensors. By this approach, our method could utilize both rich and lossless distance and speed clues from radar echoes and rich semantic clues from images, making our method surpass all existing methods on the RADIal dataset, and approach the performance of LiDAR. The code will be released on https://github.com/tusen-ai/EchoFusion.

## 1 Introduction

Robust and accurate perception is a long-standing challenge in Autonomous Driving Systems (ADS). The full perception system usually relies on multiple sensor fusion including camera, LiDAR, and radar. Camera provides rich semantic cues of objects and excellent resolution, while LiDAR is capable of capturing highly accurate spatial information. However, neither camera nor LiDAR can directly measure speed or survive in adverse weather conditions, such as fog, sandstorms, and snowstorms [26]. Although radar can overcome the aforementioned problems, even the point cloud from the latest 4D millimeter wave (mmWave) radars suffers from severe sparsity and low angular resolution, which makes it hard to discriminate objects and backgrounds solely. Fortunately, radar and camera are highly complementary to each other. Fusing these two types of sensors becomes a promising solution.

Radar data are usually represented as point clouds. Technically, the radar point cloud stems from raw Analog-Digital-Converter (ADC) data received by antennas. Range, speed, and angle of arrival (AoA) information of perceivable objects can be explicitly extracted in the frequency domain by consecutively applying FFT along corresponding dimensions. Among these steps, side lobe suppression and constant false alarm rate (CFAR) detector [40] are usually adopted to reduce the noise and false alarms. As a result, all the derived points are equipped with spatial coordinates, speed, and reflection intensity. Though these noise suppression operations can significantly reduce the data size and further computation costs, the resulting radar point cloud is extremely sparse, as shown in Figure 1. Even worse, quite a lot of useful information is lost during CFAR, which is adverse to the accurate perception of environments and subsequent fusion with other sensors [18]. Raw data serves as a

37th Conference on Neural Information Processing Systems (NeurIPS 2023).

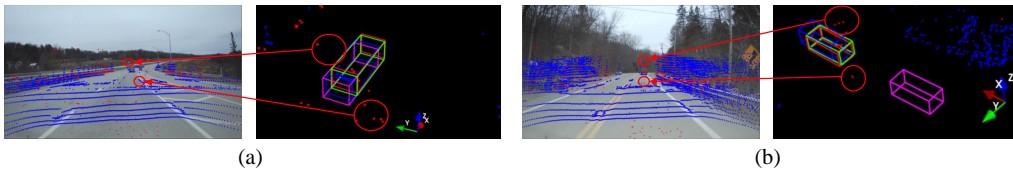

Figure 1: Illustration of sparsity and false alarms of the radar point cloud. The red points and the blue points correspond to data from radar and LiDAR, respectively. We visualize the predicted bounding boxes of different variants of our proposed method. red denotes ground-truth, purple denotes predictions from image only modality, green denotes predictions from image and radar range-time data fusion, and blue denotes predictions from image and radar point cloud fusion. Best viewed on screen.

potential solution to alleviate these problems. However, how to efficiently utilize it, especially fusing with other sensors remains an open problem.

Recently, Birds-Eye-View (BEV) based methods are prevalent for driving scenario perception. It integrates features from multiple views and multiple modalities into a unified 3D representation space [4; 17; 24]. Among the varieties of BEV-based methods, PolarFormer [9] divides the 3D space in polar coordinates, which matches the format of raw radar data better. As such, we chose PolarFormer as our baseline model.

In this work, we present EchoFusion, a method designated for fusing raw radar data with images. We observe that a set of 3D positions exactly corresponds to one column in the image and one row in raw radar data. Thereby we propose a novel polar-aligned attention (PAA) technique to efficiently fuse features from these two different modalities. With row-wise cross-attention on radar data and column-wise attention on image, PAA could precisely aggregate essential features from both modalities while maintaining simple and efficient implementation. Finally, a polar BEV decoder refines object queries for accurate bounding box predictions.

The contribution of our paper is three-folds:

1. We are the first to fuse raw radar data with images in BEV space. Specifically, we propose a novel Polar-Aligned Attention module to guide the network effectively learn radar and image features.

2. We relabel the RADIal dataset [39] with accurate 3D bounding box annotations. The new annotations is published with our codebase together.

3. Extensive experiments show that our method has outperformed all the existing methods and achieved promising BEV detection performance, even approaching the LiDAR-based method.

## 2 Related Work

### 2.1 Deep Learning on Radar

#### 2.1.1 Radar Data Representation and Datasets

Traditional radar usually only outputs sparse points [2; 5; 42] for use. However, such points are subjective to heavy signal processing methods to reduce noisy observation and alleviate bandwidth limitation. Nevertheless, such hand-crafted pipelines make the subsequent fusion with other sensors intractable. Consequently, some attempts have been made to provide upstream data, such as range-azimuth-Doppler (RAD) tensor [34; 54], range-azimuth (RA) maps [44], range-Doppler (RD) spectrums [29] or even raw Analog Digital Converter (ADC) data [19; 29]. Note that the RAD, RA, and RD tensors are generated from ADC data using several times (almost) lossless FFT. Thus we call all these four types of data "radar raw data" in the sequel.

Since our motivation is to deeply fuse with other sensors, we only consider the datasets providing data beyond point cloud. Among them, both RADIal [39], Radatron [26] and KRadar [35] provide such data. But Radatron only covers a single scenario while lacking camera calibration, LiDAR modality,

Table 1: 4D Radar Datasets Comparison. Data: PC, RA, ADC denote to point cloud, range azimuth map and raw ADC data; Sensors: C, $C_s$, L, O denote to camera, stereo camera, LiDAR, and odometer; Scenarios: U, S, H denote to urban (city), suburban, highway; Annotations: 3D, BEV, T, $P_o$, M denote to 3D bounding box, BEV bounding box, track ID, object-level point and segmentation mask; Classes and Size denote the number of classes and the number of annotated frames of each dataset.

| Dataset | Data | Other Sensors | Scenarios | Annotations | Classes | Size |
|---------|------|---------------|-----------|-------------|---------|------|
| Astyx [28] | PC | CL | SH | 3D | 7 | 500 |
| View-of-Delft [37] | PC | $C_s$LO | U | 3D,T | 13 | 8693 |
| RADIal [39] | ADC,RA,PC | CLO | USH | $P_o$,M | 1 | 8252 |
| TJ4DRadSet [57] | PC | CLO | U | 3D,T | 5 | 7757 |
| Radatron [26] | ADC,RA | $C_s$ | U | BEV | 1 | 16K |
| KRadar [35] | RA, PC | CLO | USH | 3D,T | 5 | 35K |

and height annotation, while KRadar only provides radar RA map and PCD formats. Taking the factors mentioned above into account, we carry out our experiments mainly on the RADIal dataset, and provide additional results on a subset of KRadar dataset. However, the annotations of RADIal only contain object-level points. For a more comprehensive and objective evaluation, we further annotate the bounding box size and heading of each object based on dense LiDAR points. The annotations can be found in our codebase.

### 2.1.2 Object Detection on Radar Data

Traditional radar object detection methods that utilize point cloud face challenges of super sparsity due to the low angular resolution. Practical treatments include occupancy grid mapping [59] and separate processing for static and dynamic objects [43]. 4D radar technology improves the azimuth resolution and provides additional elevation resolution, which enables the combination of static and dynamic object detection in a single network, using either PointPillars [37] or tailor-made self-attention modules [1; 51].

The pre-CFAR(Constant False Alarm Rate Detector) data provides rich information of both targets and backgrounds. RAD tensor is fully decoded but brings high demand of storage and computation. Hence, Zhang et al. [54] takes the Doppler dimension as channels and Major et al. [27] projects RAD tensor to multiple 2D views. RTCNet [36] divides RAD tensor into small cubes and applies 3D CNN to reduce computation burden. Besides, [39; 55] take complex RD spectrum as input and apply neural networks to automatically extract spatial information. Networks such as RODNet [48] adopt RA maps for detection, which avoid false alarms caused by extended Doppler profile. Despite the aforementioned research works, the utilization of ADC radar data has recently gained increasing attention within the community [7; 52]. However, their results remain unsatisfactory.

The fusion of different type of sensors provides complementary cues, leading to more robust performance. At input level, radar point clouds are usually projected as pseudo-images and concatenated with camera images [3; 33]. At Region of Interest (RoI) level, some approaches [31; 32] consecutively refine the RoIs by radar and other modalities, while others [12; 14] unify RoIs generated independently by different sensors. At feature level, [50; 56] integrate feature maps generated from different modalities, while [11; 13] use RoIs to crop and merge features across modalities. To the best of our knowledge, we are the first to deeply fuse radar using raw data with other modalities in a unified BEV perspective.

### 2.2 BEV 3D Object Detector

BEV object detection bursts out a rising wave of research due to its vast success in 3D perception tasks. One thread is to adopt transformations to project 2D image features to 3D BEV space for further detection, such as OFT [41] and LSS [38]. Another thread applies initialized BEV queries [58] or object queries [22; 23] to iteratively and automatically sample features from multi-view images. Based on these advanced techniques, BEVFusion [24] explores the advantages of BEV representation in multi-sensor fusion and achieves impressive performances. Despite that, how to make full use of other information sources also attracts the interest of researchers. BEVFormer [17] and its variant [53] utilize temporal information to enhance detection capability, while BEVStereo [16] and STS

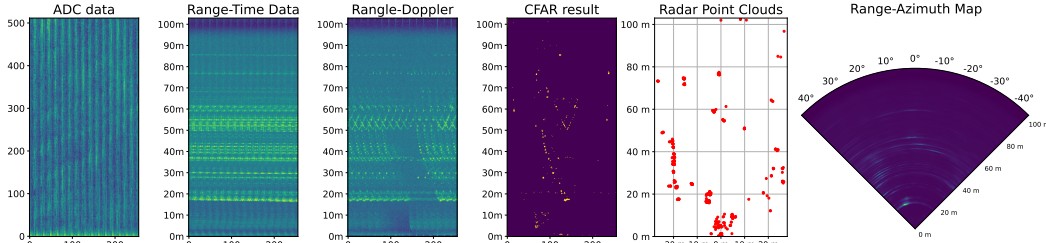

Figure 2: Radar data formats used in this paper. All the axes that indicate spatial location are labeled with units. For ADC data, the vertical unit is the sampling cycle. And for ADC data, Range-Time data, the horizontal unit is the chirp cycle. Due to Doppler domain multiplexing (DDM), the real speed is periodic on the Doppler dimension, which is hard to label (More details of DDM can be found in the appendix). We use the index number of the Doppler dimension as horizontal unit instead in Range-Doppler and CFAR result.

[49] explore how the estimated depth can benefit BEV-based detection. Besides, PolarFormer [9] introduces the polar grid setting and proves its effectiveness in environment perception, which is closely related to our work.

## 3 Preliminary

In the automotive industry, multiple-input, multiple-output (MIMO) frequency-modulated continuous-wave (FMCW) radars are adopted [59] to balance cost, size, and performance. Each transmitting antenna (Tx) transmits a sequence of FMCW waveforms, also named chirps, which will be reflected by objects and returned to the receiving antennas (Rx). Then the echo signal will be first mixed with the corresponding emitted signal and then passed through a low pass filter, so as to obtain the intermediate frequency (IF) signal. The discretely sampled IF signal, which includes the phase difference of emitted and returned signal, is called **ADC data** [46]. The ADC data has three dimensions, fast time, slow time, and channel, which have physical correspondence with range, speed, and angle respectively.

Because of the continuous-wave nature, the IF signal of a chirp contains a frequency shift caused by the time delay of traveling between radar and objects. Then the radial distance of obstacles can be extracted by applying Fast Fourier Transform (FFT) on each chirp, i.e. fast-time dimension. This FFT operation is also called range-FFT. The resulting complex signal is denoted as **range-time (RT) data**. Next, a second FFT (Doppler FFT) along different chirps, i.e. slow-time dimension, is conducted to extract the phase shift between two chirps caused by the motion of targets. The derived data is named **range-Doppler (RD) spectrum**. We denote $N_{Tx}$ and $N_{Rx}$ respectively as the number of transmitting and receiving antennas. And each grid on the RD spectrum has $N_{Tx} \times N_{Rx}$ channels. The small distance between adjacent virtual antennas leads to phase shifts among different antennas, which is dependent on the AoA. Thus a third FFT (angle-FFT) can be applied along the channel axis to unwrap AoA. The AoA can be further decoded to azimuth and elevation based on the radar antenna calibration. The final 4D tensor is referred to as the **range-elevation-azimuth-Doppler (READ) tensor**.

If point cloud is desired, the RD spectrum will be first processed by the CFAR algorithm to filter out peaks as candidate points. Then the angle-FFT will only be executed on these points. Final **radar point cloud (PCD)** consists of 3D coordinates, speed, and reflective intensity. In traditional radar signal processing, there is another data format named **range-azimuth (RA) map**, which is obtained by decoding the azimuth of a single elevation on the RD spectrum and compressing the Doppler dimension to one channel. For comprehensive analyses, we include this modality in our experiments as well. The data formats mentioned above are illustrated in Figure 2, except for READ tensor because it is difficult to visualize.

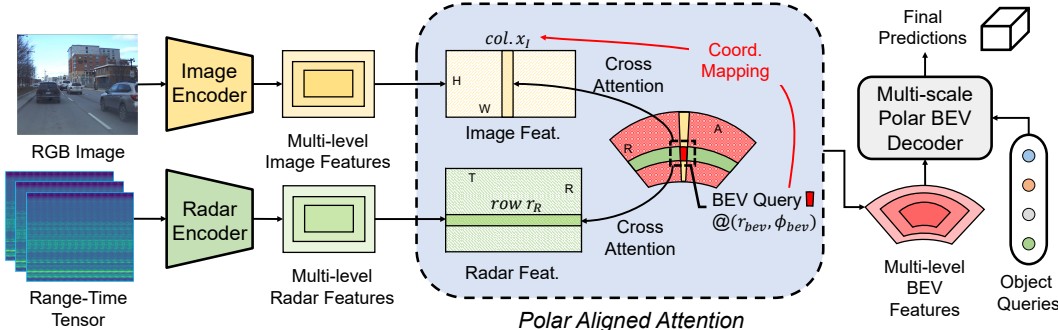

Figure 3: Overall pipeline of the proposed EchoFusion. H and W are the height and weight of the image, while R, A, and T are the bin number of range, azimuth, and Doppler dimension, respectively. The Coord. Mapping in the figure is explained in section 4.2.

## 4  Methods

In this section, we will introduce our network design. Firstly we will give a brief introduction of the whole pipeline in Section 4.1. Then our proposed Polar-Aligned Attention module will be elaborated in Section 4.2 and Section 4.3. Finally, the detection head and loss functions are introduced in Section 4.4.

### 4.1  Overall Structure

The overall architecture of our model is shown in Figure 3. First, we use separate backbones and FPNs [20] to extract multi-level features from RT radar maps and images. Then the polar queries aggregate camera features and radar features by cross-attention. Finally, the multi-scale polar BEV map will be passed through a transformer-based decoder with object queries. The final score and bounding box predictions can be obtained by decoding the output object queries. The core of our EchoFusion is how to fuse the camera and radar slow-time data. We fuse them using a novel strategy named Polar Aligned Attention (PA-Attention). In the next two subsections, we will explain this strategy in detail.

### 4.2  Column-wise Image Fusion

After obtaining image and radar features from corresponding encoders, we first initialize the $l$-th level polar queries $\boldsymbol{Q_l} \in \mathbb{R}^{R_l \times A_l \times d}$ uniformly in polar BEV space. Here, $R_l$, $A_l$, and $d$ respectively denote the shape of range, azimuth, and channel dimensions. For brevity, we omit $l$ in the following two subsections. Given a query $q \in \mathbb{R}^d$ located at $(r_{bev}, \phi_{bev})$, it corresponds to a pillar centered at $(r_{bev}, \phi_{bev})$ with infinite height. Taking the coordinate system defined in RADIal [39], $x_R$, $y_R$, and $z_R$ axes respectively points to the front, left, and upright, while subscript $R$ means radar coordinate. Similarly, the camera coordinates are denoted as $(x_C, y_C, z_C)$, which points to the right, down, and front, respectively. Thus we have correspondence in Cartesian coordinate system as follows:

$$x_R = r_{bev} \cos\left(\phi_{bev}\right), \quad y_R = r_{bev} \sin\left(\phi_{bev}\right). \tag{1}$$

Using extrinsic matrix and intrinsic matrix, we have the following formulation:

$$\frac{x_I - u_0}{f_x} = \frac{x_C}{z_C}, \quad \begin{pmatrix} x_C \\ y_C \\ z_C \end{pmatrix} = R \begin{pmatrix} x_R \\ y_R \\ z_R \end{pmatrix} + T, T = \begin{pmatrix} d_1 \\ d_2 \\ d_3 \end{pmatrix}, \tag{2}$$

where $u_0$, $f_x$ are principal point offset and focal length on $x$-axis in the intrinsic matrix, $x_I$ is the image column index that corresponds to the point $(x_R, y_R, z_R)$, while $R$ and $T$ are rotation matrix and translation vector of the calibration matrix between camera and radar, respectively. Under mild distortion conditions and normally adopted sensor setting, $x_I$ is determined by pillar position and calibration parameters as follows:

$$x_I \approx u_0 + f_x \frac{-r_{bev} \sin\phi_{bev} + d_1}{r_{bev} \cos\phi_{bev} + d_3}. \tag{3}$$

Limited by the number of pages, we leave the detailed proof in the appendix.

However, the height is not specified for a query, while for a pixel $(x_I, y_I)$ on the monocular image, its depth is unknown as well. Thus we cannot precisely associate a query to the image feature, but we can correspond queries of the same azimuth to the same column of images. Enlightened by this observation, we use a versatile cross-attention mechanism to flexibly aggregate required features from images. Namely, all the queries with the same azimuth form the query matrix, while all the features in the corresponding column in the image form the key and value matrix. Formally, these two steps can be expressed as follows:

$$
\begin{aligned}
F_I(x_I, \cdot) &= \text{Stack}\left([f_I(x_I, y_I)]\right) \in \mathbb{R}^{H_l \times d}, \quad \forall y_I \in [0, H_l - 1], \\
\hat{q}(r_{bev}, \phi_{bev}) &= \text{CrossAttn}\left(q(r_{bev}, \phi_{bev}), F_I(x_I, \cdot), F_I(x_I, \cdot)\right),
\end{aligned}
\tag{4}
$$

where $f_I(x_I, y_I)$ is the image feature located at $x_I, y_I$, while details of cross attention function CrossAttn can be found in [47].

## 4.3 Range-wise Radar Fusion

We denote updated BEV queries as $\hat{Q}_l \in \mathbb{R}^{R_l \times A_l \times d}$. After that, it will be used to sample features from radar. Since the radar feature map shares the same range partition with the BEV queries, the updated query $\hat{q}(r_{bev}, \phi_{bev})$ matches the same range index of the radar feature, i.e. $r_R = r_{bev}$. To aggregate all the features from all chirps in one frame, we perform cross attention for a certain query $\hat{q}$ with the row $r_R$ in RT map as key and value. Formally,

$$
\begin{aligned}
F_R(r_R, \cdot) &= \text{Stack}\left([f_R(r_R, t_R)]\right) \in \mathbb{R}^{R_l \times d}, \quad \forall t_R \in [0, T_l - 1], \\
\tilde{q}(r_{bev}, \phi_{bev}) &= \text{CrossAttn}\left(\hat{q}(r_{bev}, \phi_{bev}), F_R(r_R, \cdot), F_R(r_R, \cdot)\right),
\end{aligned}
\tag{5}
$$

where $R_l$ and $T_l$ are the height and width of the radar feature map, and $f_R(r_R, t_R) \in \mathbb{R}^d$ is the radar feature of position $(r_R, t_R)$.

Note that though we don't explicitly decode AoA as in RA map, the AoA has been implicitly encoded in the phase difference of the response of different virtual antenna. Consequently, it can be learned from the features in RT map indirectly.

## 4.4 Head and Loss

We follow polar BEV decoders [9] to decode multi-level BEV features and make predictions from object queries. Since we don't have velocity annotation on this dataset, we remove the corresponding branch. The regression targets include $(d_\rho, d_\phi, d_z, \log l, \log w, \log h, \sin(\theta_{ori} - \phi), \cos(\theta_{ori} - \phi))$, where $d_\rho, d_\phi, d_z$ are relative offset to the reference point $(\rho, \phi, z)$, $l, w, h, \theta_{ori}$ are length, width, height, and orientation of bounding box. The classification and regression tasks are respectively supervised by Focal loss [21] and L1 loss.

# 5 Experiments

In this section, we conduct thorough experiments to validate the effectiveness of our approach. We first introduce the dataset and metrics used in our experiments, then followed by comparisons with other state-of-the-art methods. Lastly, we ablate some designs and potential input format of raw radar data in our method, and discuss some limitations.

## 5.1 Experimental Settings

**The RADIal Dataset** [39] provides 8252 annotated frames with synchronized multi-sensor data. Training, validation, and test sets contain 6231, 986, and 1035 frames, respectively. Each object is annotated with a 3D center coordinate of the visible face of the vehicle. The size and orientation of the objects are pre-defined templates, i.e. they are the same for all objects. In the evaluation, precisions and recalls under ten thresholds are calculated. The average precision (AP), average recall (AR), and F1 score are obtained through the ten precisions and ten recalls as the evaluation metric. Range error (RE) and azimuth error (AE) are also reported to evaluate localization accuracy. The annotation method and metrics are quite different to those in common autonomous driving datasets

Table 2: Detection performances on RADIal Dataset test split with original protocol [39]. C, RD, and RT respectively refer to camera, range-Doppler spectrum, and range-time data. Our method achieves the best performance in both average precision (AP), average recall (AR), and F1-score (F1). The best result of each metric is in **bold**.

| Methods | Modality | AP(%)↑ | AR(%)↑ | F1(%)↑ | RE(m)↓ | AE(°)↓ |
|---------|----------|--------|--------|--------|--------|--------|
| FFTRadNet [39] | RD | 96.84 | 82.18 | 88.91 | **0.11** | 0.17 |
| T-FFTRadNet [7] | ADC | 89.60 | 89.50 | 89.50 | 0.15 | 0.12 |
| ADCNet [52] | ADC | 95.00 | 89.00 | 91.90 | 0.13 | **0.11** |
| CMS [10] | RD&C | 96.90 | 83.50 | 89.70 | 0.45 | n/a |
| EchoFusion | RT&C | **96.95** | **93.43** | **95.16** | 0.12 | 0.18 |

[2; 45]. Nonetheless, to compare with state-of-the-art methods, we still list our results using the metrics defined in RADIal [39] as a reference.

**Re-annotation and more metrics.** Based on the provided LiDAR point cloud, we have refined the cluster-center-based annotation to 3D bounding box form, which includes center position, scale, and heading in 3D space. Capitalizing on these new annotations, we are able to apply metrics that are vastly used in 3D object detection tasks. Specifically, we take AP defined in Waymo dataset [45] as our main metrics. Since the LiDAR used in RADIal is only 16-beam, we cannot label accurate height for the objects. So we mainly utilize BEV IoU for evaluation. We use the threshold of 0.7 for normal IoU and longitudinal error-tolerant (LET) IoU computation [8]. Considering the spatial distribution of ground-truth, the distance evaluation is further broken down into two categories: 0 to 50 meters, 50 to 100 meters. For a more comprehensive evaluation, we also introduce the error metrics defined in nuScenes dataset [2], including ATE, ASE, and AOE for measuring errors of translation, scale, and orientation.

**The KRadar Dataset** [35] is a recently published 4D radar dataset, which contains 58 sequences, i.e. 35K frames of synchronized multi-modality data. However, we have limited time to obtain the full data via shipping, so we only provide results on the sub-dataset stored on Google Drive, which consists of 20 sequences. This so-called KRadar-20 dataset includes 6493 training samples and 6515 test samples, of which 50 % are nighttime data. The training samples are named trainval split, it is further split into train split of 5190 samples and val split of 1303 samples for ablation. Metrics follow KITTI [6] protocols with 40 recall positions. Different from RADIal, it only contains radar formats of range-azimuth map (RA map) and radar point cloud. But the RA map of KRadar contains an extra height dimension, and it enables us to test 3D prediction ability with 4D radar data.

**Radar Formats.** As introduced in Section 3, there are multiple formats of radar raw data, such as ADC, RT, RD, READ. These formats can be converted sequentially by FFT, which is a linear operation that can be absorbed into the linear layers. So from the perspective of neural network, these formats are equivalent. However, our proposed Polar-Aligned Attention requires a range dimension for indexing key and value in cross-attention, so we choose RT as a representative in the experiments. We also conduct experiments on RA to investigate the necessity of explicit azimuth.

**Implementation Details.** All our experiments are carried out on eight 3090 GPUs with the same batch size of 8. For the image backbone, we adopt ResNet-50 with pre-trained weights provided by BEVFormer [17]. For LiDAR and radar point cloud branch, we borrow the voxel encoder from PointPillar [15] and use ResNet-50 with modified strides as the backbone. Other representations of radar data share similar modified ResNet-50 as the backbone, but with an extra $1 \times 1$ convolution at the beginning for channel number alignment. To prevent overfitting, the image-only variant is trained for only 12 epochs while others are trained for 24 epochs. AdamW [25] is adopted as the optimizer and a learning rate of 5e-5 is shared for all models. Due to the limited size of the dataset, we find that the results are unstable across different runs. To make a fair comparison, each experiment is repeated three times. In modality ablation, we show the mean and variance of each experiment result.

Table 3: BEV detection performances on RADIal Dataset test split with refined 3D ground-truth bounding box. C, RD, and RT respectively refer to camera, range-Doppler spectrum, and range-time data. The best result of each metric is in bold. Our method exceeds FFTRadNet by a large margin.

| Methods | Modality | BEV AP@0.7(%)↑ | | | LET-BEV-AP@0.7(%)↑ | | |
| | | Overall | 0 - 50m | 50 - 100m | Overall | 0 - 50m | 50 - 100m |
| --- | --- | --- | --- | --- | --- | --- | --- |
| FFTRadNet3D[39] | RD | 57.26 | 68.66 | 52.28 | 62.81 | 75.55 | 57.98 |
| EchoFusion | RT&C | **84.92** | **87.56** | **91.06** | **88.86** | **92.81** | **94.81** |

Table 4: BEV detection performances on KRadar Dataset test split with KITTI protocol. 0.3, 0.5, and 0.7 are IoU thresholds. The inputs of EchoFusion are image and range-azimuth map. The best result of each metric is in **bold**.

| Training Set | Method | BEV AP(%)↑ | | | 3D AP(%)↑ | | |
| | | AP@0.3 | AP@0.5 | AP@0.7 | AP@0.3 | AP@0.5 | AP@0.7 |
| --- | --- | --- | --- | --- | --- | --- | --- |
| KRadar | RTNH[35] | 58.04 | 42.60 | 10.69 | 49.65 | 17.87 | 0.45 |
| KRadar-20-trainval | RTNH[35] | 61.38 | 46.47 | 10.47 | 53.05 | 17.98 | 3.03 |
| KRadar-20-trainval | EchoFusion | **69.95** | **57.28** | **33.07** | **68.35** | **43.87** | **14.00** |

## 5.2 Comparison to State-of-the-art Methods

### 5.2.1 Results on RADIal Dataset

We first compare our method with state-of-the-art detectors on AP, AR, range error, and azimuth error defined in RADIal [39], with original object-level point annotations[1]. The results are illustrated in Table 2. Our method significantly improves AR and F1 performance and achieves a remarkable improvement over all existing detectors including CMS [10], which also integrates both radar raw data and camera data. Note that RE and AE are not quite informative because these two metrics are calculated on all the recalled objects of a method, which is unfair to high-recall models like ours.

To evaluate in a more rigorous and informative way, we also conduct experiments on the refined annotations and vastly adopted metrics in 3D object detection. FFTRadNet [39] is lifted up with additional branches to predict the scale and orientation of targets. We can only compare with this FFTRadNet variant since codes of other algorithms [7; 10; 52] are not available yet. The results in Table 3 show that the proposed method outperforms FFTRadNet by a large margin. This significant improvement reveals the huge benefit of unleashing the power of radar data in multi-modality fusion.

### 5.2.2 Results on KRadar Dataset

The baseline on KRadar dataset is the official RTNH [35], which requires radar point cloud. Here, the proposed EchoFusion is input with image and RA map. To align with the baseline, we take the radius and azimuth range respectively as [0, 72] m and [-20, 20] degrees by the limit of camera field of view. The results are shown in Table 4. With RA map and image fusion, our EchoFusion improves over 10 points at each metric, except for BEV AP of IoU threshold 0.3. And since KRadar has higher vertical resolution and more accurate 3D groundtruth than RADIal, our method achieves more considerable improvements in 3D metrics.

## 5.3 Ablation Studies

We ablate the input format to the radar feature extraction network in Table 5. It shows that the decomposition of complex input can improve overall detection performance, especially for long-range perception. The magnitude and phase representation encodes a non-linear transformation of the complex representation, which may relate to the final prediction target better. We also tried to remove the pre-training of the image branch. The result drops significantly, which confirms that in this relatively small dataset, a good pre-training is crucial for good performance.

---

[1]The official implementation of these metrics are different from common implementations. Specifically, AP and AR are averaged on score thresholds from 0.1 to 0.9 with step of 0.1 and IoU threshold of 0.5. And F1 score is directly calculated from AP and AR defined above.

Table 5: Ablation studies on our EchoFusion. Complex means interpreting input features as real and imaginary (IQ) or magnitude and phase (MP). Pretrain means whether to use the pre-trained image backbone. The best result of each metric is in **bold**.

| Conditions | | BEV AP@0.7(%)↑ | | | LET-BEV-AP@0.7(%)↑ | | |
|---|---|---|---|---|---|---|---|
| Complex | Pretrain | Overall | 0 - 50m | 50 - 100m | Overall | 0 - 50m | 50 - 100m |
| MP | w | **84.92** | **87.56** | **91.06** | **88.86** | **92.81** | **94.81** |
| IQ | w | 82.64 | 87.09 | 87.81 | 88.06 | 92.25 | 93.24 |
| MP | w/o | 74.34 | 81.26 | 78.71 | 78.90 | 85.44 | 84.28 |

Table 6: BEV detection performances of different modality combinations of our algorithm on RADIal Dataset test split. Refined 3D bounding box annotations are applied. C, L, RD, RT, RA, RP respectively refer to the camera, LiDAR, range-Doppler spectrum, range-time data, range-azimuth map, and radar point cloud. The best result without LiDAR of each metric are in **bold**, and the results with LiDAR are on gray background.

| C | RT | RA | RP | L | Overall | 0 - 50m | 50 - 100m | Overall | 0 - 50m | 50 - 100m |
|---|---|---|---|---|---|---|---|---|---|---|
| | | | | | BEV AP@0.7(%)↑ | | | LET-BEV-AP@0.7(%)↑ | | |
| ✓ | | | | | 10.07±1.10 | 21.68±3.11 | 2.21±0.59 | 56.92±3.91 | **78.98±1.41** | 36.20±5.81 |
| | ✓ | | | | 48.26±0.92 | 62.33±4.00 | 45.05±0.98 | 55.04±1.25 | 68.56±4.11 | 52.30±1.82 |
| | | ✓ | | | 50.67±1.03 | 65.05±1.59 | 42.98±0.51 | 58.53±0.13 | 72.95±0.78 | 51.41±0.54 |
| | | | ✓ | | **53.55±0.70** | **66.19±1.72** | **47.41±1.06** | **62.59±1.14** | 76.45±2.06 | **56.73±1.33** |
| | | | | ✓ | 84.47±0.15 | 86.92±0.61 | 91.91±0.13 | 86.07±0.44 | 88.55±0.79 | 93.17±0.10 |
| ✓ | ✓ | | | | **84.92±0.98** | 87.56±1.58 | 91.06±1.18 | 88.86±0.62 | 92.81±1.37 | 94.81±0.52 |
| ✓ | | ✓ | | | 84.77±0.65 | **87.93±0.60** | **91.48±0.28** | **89.54±0.54** | **93.83±0.56** | **94.87±0.58** |
| ✓ | | | ✓ | | 82.35±0.93 | 86.97±1.01 | 86.39±0.74 | 88.35±0.78 | 93.07±0.80 | 92.77±0.69 |
| ✓ | | | | ✓ | 86.35±1.15 | 88.81±1.40 | 94.25±1.68 | 88.44±0.79 | 91.63±1.34 | 95.31±1.28 |

## 5.4 Comparison of Different Modalities

To further study the effects of different modalities, we change the input format indicated in Figure 2 and corresponding backbones, and test with and without image modality. Table 6 presents the results, more results on KRadar can be found in our appendix, and our findings are listed as follows.

**Comparing single-modality results** It is not surprising that LiDAR ranks first while the camera is the worst in terms of AP. Though poor in depth estimation, the camera has excellent angular resolution, which is beneficial for LET-BEV-AP. It achieves comparable results with radar in LET-BEV-AP. Besides, without the guidance of image, the radar-only network gets better performance as more hand-crafted post-processing are involved. However, their metrics still lag far from those of LiDAR.

**Comparing multi-modality with single modality** The power of radar data is released by fusing with image. No matter in what data format, all the radar raw representations gain around 30 points improvement by fusing with images. By combining image and range-time data, we obtain BEV AP that is only 0.03 less than that of LiDAR only method. And the LET-BEV-AP and long-range detection ability are even better. We argue that it is mainly because images provide enough clues to decode the essential information from raw radar representation.

**Comparing multi-modality results** When integrating with images, the LiDAR-fused method still outperforms other radar-fused methods. In terms of different representations of radar data, both RT and RA outperform traditional cloud points, especially in the 50-100m range, in which the difference is as large as 4 points in AP. We owe this finding to the information loss and false alarm of radar point cloud as shown in Figure 1. The performance gap between RT and RA with camera modality is within the error bar, indicating that it is not necessary to explicitly solve the azimuth, nor necessary to permute the Doppler-angle dimensions as FFTRadNet [39] does.

## 5.5 Discussion and Limitation

Firstly, it is worth noticing that although the overall performance of radar is improved by fusing with the camera, the performance in the short-range is lower than that in the long-range. We speculate the main reason is inaccurate annotation. The 16-beam LiDAR provided in RADIal is too sparse at

Table 7: 3D detection performances on RADIal Dataset test split with refined 3D ground-truth bounding box. C, L, and RT respectively refer to camera, LiDAR, and radar-time data. The best result of each metric is in bold.

| | 3D AP@0.5(%)↑ | | | Average Error@0.5↓ | | |
|---|---|---|---|---|---|---|
| Modality | Overall | 0 - 50m | 50 - 100m | ATE | ASE | AOE |
| L | 62.69 | 80.04 | 54.26 | 0.243 | **0.171** | 0.027 |
| L&C | **68.36** | **84.74** | **61.54** | **0.239** | 0.180 | **0.020** |
| RT&C | 39.81 | 56.05 | 31.24 | 0.301 | 0.173 | **0.020** |

a farther distance for annotators to give an accurate size for the long-range objects. In such cases, they are required to assign a template with a fixed size to these objects. These annotations make the problem much easier for farther distances, which leads to higher performance than near distances.

Secondly, though the radial speed label is included in the original annotations of RADIal, radial speed is not quite useful for downstream modules like planning since they need accurate longitudinal and lateral velocity to predict whether the vehicles will interfere with the ego vehicle. However, the synchronized frames are not consecutive in this dataset, it is hard to label velocities for the objects. Considering these factors, velocity prediction is not included in our task.

Finally, as shown in Table 7, our method still lags far from LiDAR-based methods. The primary limitation can be attributed to imperfect z localization, resulting in relatively high Average Localization Error (ATE). The inferior performance is mainly due to coarse LiDAR annotation and inferior radar elevation resolution. The low elevation distinguishing power of LiDAR makes annotation error much worse at long range. As a result, both LiDAR-based and radar-based methods experience a significant drop in 3D AP within the 50-100m interval. But when equipped with high elevation resolution and accurate 3D groundtruth provided by KRadar, our method shows excellent performance in 3D metrics, as shown in Table 4. This emphasizes the pressing need for a raw radar dataset of higher quality that enables better exploration of radar data usage.

## 6 Conclusion and Future Work

In this work, we have proposed a novel method for radar raw data fusion with other sensors in a BEV space. Our proposed EchoFusion is concise and effective, which outperforms previous work by a significant margin. We are the first to demonstrate the potential of radar as a low-cost alternative for LiDAR in autonomous driving systems through thorough analyses and experiments.

This work is only a starting point to study how raw radar data can be exploited. However, many attempts are limited by the available datasets. We urge a large-scale and high-quality dataset. However, the acquisition of high-quality multi-modality data with accurate annotation needs great effort and deliberate design for clock synchronization and high storage demand. We will try to build the dataset to facilitate further research.

**Societal Impacts** Our method can be deployed in the autonomous driving system. Performance loss caused by improper usage may increase security risks.

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

# Supplementary Material

## A    Proof for column and pillar correspondence

In this section, we will provide a detailed proof for the correspondence between pillar in radar coordinate and column in camera coordinate as described in Section 4.2 of our main paper.

Suppose that the rotation matrix and translation vector between radar and camera are:

$$R = \begin{pmatrix} a_1 & b_1 & c_1 \\ a_2 & b_2 & c_2 \\ a_3 & b_3 & c_3 \end{pmatrix}, T = \begin{pmatrix} d_1 \\ d_2 \\ d_3 \end{pmatrix}. \tag{S1}$$

Then for a point located at $(r_{bev}, \phi_{bev}, z)$ under radar's polar coordinate, it has cartesian coordinate $(x_R, y_R, z_R)$:

$$x_R = r_{bev} \cos(\phi_{bev}), y_R = r_{bev} \sin(\phi_{bev}), z_R = z, \tag{S2}$$

and it can be projected to the camera frame using extrinsic parameters:

$$\begin{pmatrix} x_C \\ y_C \\ z_C \end{pmatrix} = R \begin{pmatrix} x_R \\ y_R \\ z_R \end{pmatrix} + T = \begin{pmatrix} a_1 r_{bev} \cos \phi_{bev} + b_1 r_{bev} \sin \phi_{bev} + c_1 z + d_1 \\ a_2 r_{bev} \cos \phi_{bev} + b_2 r_{bev} \sin \phi_{bev} + c_2 z + d_2 \\ a_3 r_{bev} \cos \phi_{bev} + b_3 r_{bev} \sin \phi_{bev} + c_3 z + d_3 \end{pmatrix}. \tag{S3}$$

Using the intrinsic parameters, the camera coordinate $(x_C, y_C, z_C)$ can be correlated with image pixel position $(x_I, y_I)$ using:

$$\frac{x_I - u_0}{f_x} = \frac{x_C}{z_C}, \frac{y_I - v_0}{f_y} = \frac{y_C}{z_C}, \tag{S4}$$

where $f_x$, $f_y$, $u_0$, and $v_0$ are intrinsic parameters. Our goal is to find a situation that the pillar is projected as a column on the image plane. Under this condition, $x_I$ should be irrelevant with $z$, i.e. in the equation below:

$$\frac{x_I - u_0}{f_x} = \frac{x_C}{z_C} = \frac{a_1 r_{bev} \cos \phi_{bev} + b_1 r_{bev} \sin \phi_{bev} + c_1 z + d_1}{a_3 r_{bev} \cos \phi_{bev} + b_3 r_{bev} \sin \phi_{bev} + c_3 z + d_3}. \tag{S5}$$

The coefficients of z should be 0, which means by combining the relationship between rotation matrix $R$ with roll $\alpha$, pitch $\beta$, and yaw $\gamma$ [30], we can derive:

$$\begin{cases} c_1 = \sin \gamma \sin \alpha + \cos \gamma \sin \beta \cos \alpha = 0 \\ c_3 = \cos \beta \cos \alpha = 0. \end{cases} \tag{S6}$$

Considering second row of Eq. (S6), there are two solutions $\beta = \pm \frac{\pi}{2}$ or $\alpha = \pm \frac{\pi}{2}$. Considering the first solution $\beta = \pm \frac{\pi}{2}$, the first row of Eq. (S6) is transformed to:

$$\sin \gamma \sin \alpha \pm \cos \gamma \cos \alpha = 0, \tag{S7}$$

which means:

$$\cos(\gamma \mp \alpha) = 0. \tag{S8}$$

Thus we have:

$$\beta = \frac{\pi}{2}, \gamma = \alpha \pm \frac{\pi}{2}, \alpha \in [-\pi, \pi] ,$$
$$or, \ \beta = -\frac{\pi}{2}, \gamma = -\alpha \pm \frac{\pi}{2}, \alpha \in [-\pi, \pi]. \tag{S9}$$

For another solution $\alpha = \pm \frac{\pi}{2}$, the first row of Eq. (S6) is transformed to:

$$\sin \gamma \sin \alpha + \cos \gamma \sin \beta \cos \alpha = \pm \sin \gamma = 0. \tag{S10}$$

Thus we have:

$$\alpha = \pm \frac{\pi}{2}, \gamma = 0 \ or \ \pi, \beta \in [-\pi, \pi]. \tag{S11}$$

As a result, the final solution is:

$$\beta = \frac{\pi}{2}, \gamma = \alpha \pm \frac{\pi}{2}, \alpha \in [-\pi, \pi],$$
$$or, \beta = -\frac{\pi}{2}, \gamma = -\alpha \pm \frac{\pi}{2}, \alpha \in [-\pi, \pi], \tag{S12}$$
$$or, \alpha = \pm \frac{\pi}{2}, \gamma = 0 \ or \ \pi, \beta \in [-\pi, \pi].$$

In standard ADS, the LiDAR or radar coordinate system usually has a z-axis pointing up, while the camera coordinate system has a y-axis pointing up or down. So a widely adopted practice first exchanges the y and z axes, which means a roll $\alpha = \pm\frac{\pi}{2}$. After that, a pitch $\beta$ around the vertical y-axis is executed, while further rotation around the z-axis is not required, as shown in Figure S1. In other words, the roll $\beta$ can be any value between $-\pi$ and $\pi$, while yaw $\gamma$ is set to 0. This is exactly the situation of the third solution of Eq. (S12). And under this condition, we have expression of $R$ according to [30]:

Figure S1: Illustration of coordinate system transformation.

$$R = \begin{pmatrix} \cos\beta & \pm\sin\beta & 0 \\ 0 & 0 & \mp 1 \\ -\sin\beta & \pm\cos\beta & 0 \end{pmatrix}, T = \begin{pmatrix} d_1 \\ d_2 \\ d_3 \end{pmatrix}. \quad (S13)$$

Thus, using Eq. (S3) and Eq. (S5), we can get the expression of column index $x_I$. If $\alpha = \frac{\pi}{2}$, we have:

$$x_I = u_0 + f_x \frac{r_{bev}\cos(\beta - \phi_{bev}) + d_1}{-r_{bev}\sin(\beta - \phi_{bev}) + d_3}. \quad (S14)$$

If $\alpha = -\frac{\pi}{2}$, we have:

$$x_I = u_0 + f_x \frac{r_{bev}\cos(\beta + \phi_{bev}) + d_1}{-r_{bev}\sin(\beta + \phi_{bev}) + d_3}. \quad (S15)$$

For RADIal, the rotation matrix from the radar coordinate system to the camera coordinate system is:

$$R = \begin{pmatrix} 0.0465 & -0.9989 & -0.0051 \\ -0.0476 & 0.0029 & -0.9989 \\ 0.9978 & 0.0467 & -0.0474 \end{pmatrix}, \quad (S16)$$

which is the approximation of $(\alpha, \beta, \gamma) = (\pi/2, -\pi/2, 0)$. And thus we approximately have:

$$x_I \approx u_0 + f_x \frac{-r_{bev}\sin\phi_{bev} + d_1}{r_{bev}\cos\phi_{bev} + d_3}. \quad (S17)$$

It can be inferred that $x_I$ is almost irrelevant to $z$, which means one pillar in the polar coordinate of radar corresponds to a column in the image.

## B  Visualization and Dataset Statistics

To better illustrate the performance of our method, we visualize the predictions and ground truths in Figure S2. As can be seen from the cases in the first and the second column, our method can accurately predict the position of vehicles. But there are several cases in which the accurate prediction does not get high confidence, as shown in the upper case in the third column. On the other hand, the orientation estimation error may be significant when the yaw of the vehicle is large, shown in the inferior case in the third column. The main reason is that the samples with large yaw are relatively rare in the dataset, as shown in Figure S3 (b) and (c).

As shown in Figure S3 (a), the samples of test data are rather rare within 30 meters and out of 80 meters. To comprehensively explore the detection performance of objects, we divide the range into 0-50m and 50-100m, which correspond to short-range and long-range detection performance respectively.

We further visualize scenarios that are corrected by our new labels. It is worth noting that the original annotation and prediction of previous methods are centers of objects' visible faces, and they assign a fixed scale and zero orientation when calculating Average Precision (AP). But vehicles of large angles and scales are normal in daily scenarios, as shown in Figure S4. Our new annotations based on dense LiDAR points correct this, helping a more comprehensive evaluation of existing methods. We also annotate other traffic participants such as pedestrians and cyclists. But due to their limited numbers (4 cyclists, 23 pedestrians in training split, and 17 cyclists, 0 pedestrians in test split), we don't add more fine-grained class labels.

Table S1: LET-BEV-AP results with different longitudinal tolerance $T_l$. The performances of different modality combinations of our algorithm on RADIal Dataset test split are presented, using refined 3D bounding box annotations. C, L, and RT respectively refer to the camera, LiDAR, and range-time data. The results with LiDAR are on gray background.

| Modality | | | $T_l = 0.1$, LET-BEV-AP@0.7(%)↑ | | | $T_l = 0.3$, LET-BEV-AP@0.7(%)↑ | | |
|---|---|---|---|---|---|---|---|---|
| C | RT | L | Overall | 0 - 50m | 50 - 100m | Overall | 0 - 50m | 50 - 100m |
| ✓ | | | 56.92±3.91 | 78.98±1.41 | 36.20±5.81 | 81.36±1.15 | 83.30±2.01 | 77.00±0.43 |
| | ✓ | | 55.04±1.25 | 68.56±4.11 | 52.30±1.82 | 55.34±1.91 | 69.79±4.59 | 51.02±0.07 |
| | | ✓ | 86.07±0.44 | 88.55±0.79 | 93.17±0.10 | 86.41±0.47 | 89.72±0.03 | 93.22±0.04 |
| ✓ | ✓ | | 88.86±0.62 | 92.81±1.37 | 94.81±0.52 | 88.56±0.53 | 92.58±0.70 | 94.57±0.29 |

Table S2: Recall with different IoU thresholds. The performances of different modality combinations of our algorithm on RADIal Dataset test split are presented, using refined 3D bounding box annotations. C, L, and RT respectively refer to the camera, LiDAR, and range-time data. The results with LiDAR are on gray background.

| Modality | | | BEV recall@0.7(%)↑ | | | BEV recall@0.3(%)↑ | | |
|---|---|---|---|---|---|---|---|---|
| C | RT | L | Overall | 0 - 50m | 50 - 100m | Overall | 0 - 50m | 50 - 100m |
| | ✓ | | 62.73±1.24 | 73.89±2.11 | 61.21±0.82 | 86.27±0.43 | 92.13±0.71 | 93.40±0.54 |
| | | ✓ | 86.38±0.00 | 89.11±0.08 | 93.35±0.08 | 91.40±0.08 | 96.13±0.17 | 96.95±0.16 |
| ✓ | ✓ | | 86.18±0.13 | 89.13±0.6 | 92.53±0.56 | 91.78±0.85 | 95.82±0.50 | 97.67±1.05 |

Table S3: BEV detection performances of different modality combinations of our algorithm on RADIal Dataset test split. Refined 3D bounding box annotations are applied. C, L, ADC, RD, RT, RA, RP respectively refer to the camera, LiDAR, ADC data, range-Doppler spectrum, range-time data, range-azimuth map, and radar point cloud. The best result without LiDAR of each metric are in **bold**, and the results with LiDAR are on gray background.

| Modality | | | | | | | BEV AP@0.7(%)↑ | | | LET-BEV-AP@0.7(%)↑ | | |
|---|---|---|---|---|---|---|---|---|---|---|---|---|
| C | ADC | RT | RD | RA | RP | L | Overall | 0 - 50m | 50 - 100m | Overall | 0 - 50m | 50 - 100m |
| ✓ | | | | | | | 10.07±1.10 | 21.68±3.11 | 2.21±0.59 | 56.92±3.91 | **78.98±1.41** | 36.20±5.81 |
| | ✓ | | | | | | 49.64±0.43 | 61.37±1.57 | 47.92±1.48 | 56.52±2.36 | 68.93±1.87 | 54.68±1.76 |
| | | ✓ | | | | | 48.26±0.92 | 62.33±4.00 | 45.05±0.98 | 55.04±1.25 | 68.56±4.11 | 52.30±1.82 |
| | | | ✓ | | | | 47.34±0.73 | 59.31±1.64 | 42.53±1.98 | 54.96±0.67 | 67.02±1.67 | 50.59±0.26 |
| | | | | ✓ | | | 50.67±1.03 | 65.05±1.59 | 42.98±0.51 | 58.53±0.13 | 72.95±0.78 | 51.41±0.54 |
| | | | | | ✓ | | **53.55±0.70** | **66.19±1.72** | **47.41±1.06** | **62.59±1.14** | 76.45±2.06 | **56.73±1.33** |
| | | | | | | ✓ | 84.47±0.15 | 86.92±0.61 | 91.91±0.13 | 86.07±0.44 | 88.55±0.79 | 93.17±0.10 |
| ✓ | ✓ | | | | | | 84.85±0.22 | 87.68±1.54 | 91.46±1.08 | 88.66±0.63 | 92.43±1.15 | 94.99±0.66 |
| ✓ | | ✓ | | | | | **84.92±0.98** | 87.56±1.58 | 91.06±1.18 | 88.86±0.62 | 92.81±1.37 | 94.81±0.52 |
| ✓ | | | ✓ | | | | 83.31±0.39 | 87.26±0.56 | 89.16±0.44 | 88.24±0.06 | 92.18±0.43 | 93.26±0.73 |
| ✓ | | | | ✓ | | | 84.77±0.65 | **87.93±0.60** | **91.48±0.28** | **89.54±0.54** | **93.83±0.56** | **94.87±0.58** |
| ✓ | | | | | ✓ | | 82.35±0.93 | 86.97±1.01 | 86.39±0.74 | 88.35±0.78 | 93.07±0.80 | 92.77±0.69 |
| ✓ | | | | | | ✓ | 86.35±1.15 | 88.81±1.40 | 94.25±1.68 | 88.44±0.79 | 91.63±1.34 | 95.31±1.28 |

Table S4: Ablation on the coordinate system. Experiments are carried on RADIal dataset, with RT data and image as input.

| | BEV AP@0.7(%)↑ | | | LET-BEV-AP@0.7(%)↑ | | |
|---|---|---|---|---|---|---|
| Coordinate | Overall | 0 - 50m | 50 - 100m | Overall | 0 - 50m | 50 - 100m |
| Polar | 84.92±0.98 | 87.56±1.58 | 91.06±1.18 | 88.86±0.62 | 92.81±1.37 | 94.81±0.52 |
| Sphere | 85.02±0.60 | 88.02±0.59 | 91.76±0.98 | 89.97±0.61 | 93.35±0.32 | 96.41±1.21 |

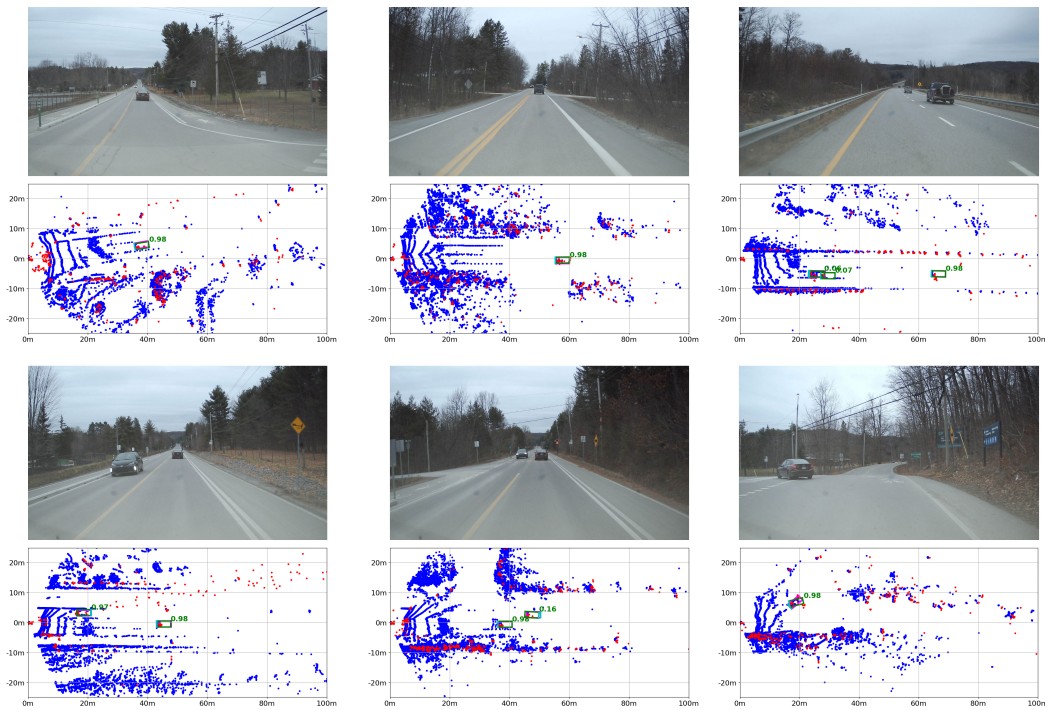

Figure S2: Visualization of ground truths and predictions from RT and image fusion. The ground truth bounding boxes are in pink, while the predicted bounding boxes are in green with the confidence score on its upper right. The LiDAR points and the radar points are respectively in blue and red. The rear edge of each car is in cyan. We recommend readers to zoom in for better detail.

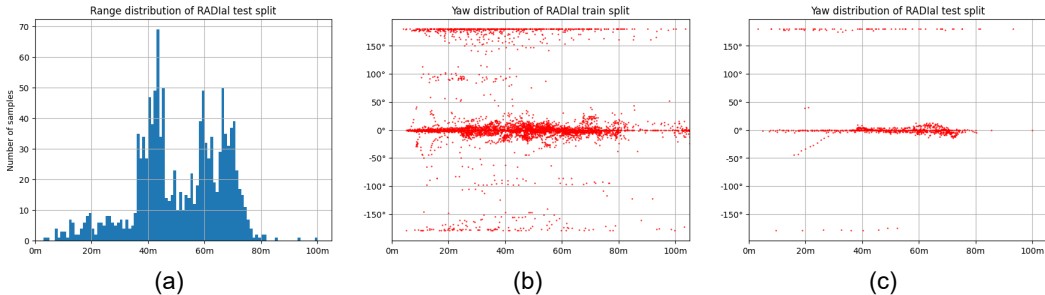

| (a) | (b) | (c) |

Figure S3: Statistics of RADIal dataset. Image (a) corresponds to sample distribution over range, while images (b) and (c) correspond to yaw distribution over the range of train and test data splits.

## C   Complementary property of radar and camera

To highlight the complementary nature of radar and camera properties, we investigated the performance of LET-BEV-AP with varying longitudinal tolerance ($T_l$), as presented in Table S1. Notably, when using a larger $T_l$ of 0.3, the camera's performance experiences a significant improvement. This suggests that the relatively lower performance of long-range LET-BEV-AP under $T_l$ of 0.1 primarily stems from depth errors. By reducing the impact of longitudinal estimation, the camera's exceptional angular perception accuracy becomes more pronounced. Conversely, the LET-BEV-AP performance of the radar-only variant remains relatively unchanged. This finding indicates that the radar's depth estimation is quite accurate and minimally affected by the value of $T_l$. Similarly, the highly accurate LiDAR-only variant and camera and RT fusion variant don't suffer from performance fluctuation as well, which supports our conclusion.

To further investigate the primary factor affecting radar performance, we conducted a comparison of BEV recall at IoU thresholds of 0.7 and 0.3, as outlined in Table S2. It can be deduced that by using a

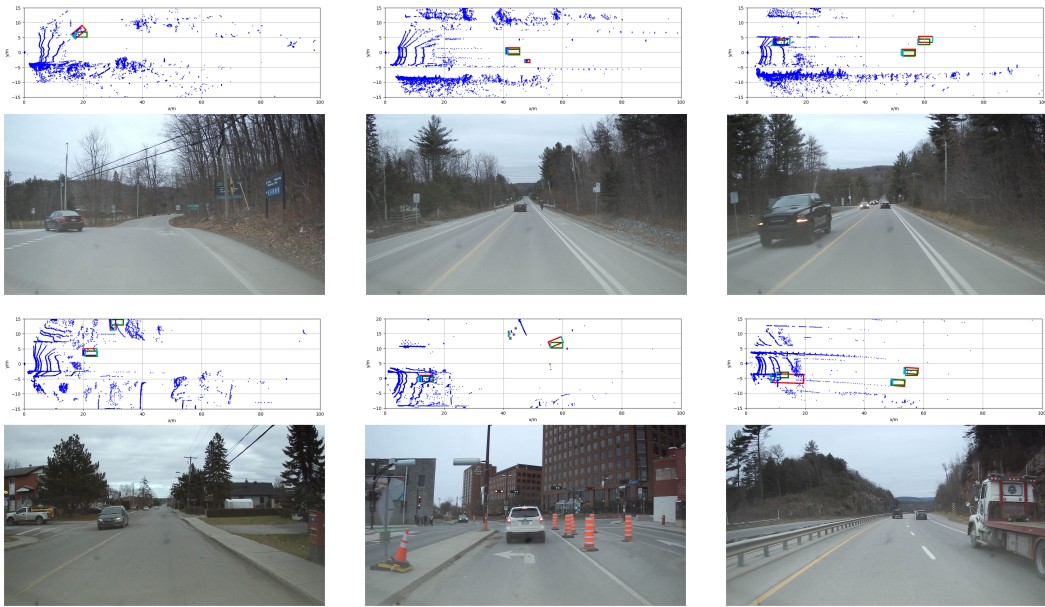

Figure S4: The visualization of the old labels and new labels under Bird's Eye View. The old labels include 3D center points and fixed scale and orientation, denoted as green boxes. The new labels are denoted as red boxes, which we offered real scale and orientation, as well as objects' accurate center position. The rear edge of each bounding box is colored in cyan. The left, middle, and right column respectively denotes the situation of large heading angle, missed traffic participants, and vehicles of large scale. We recommend readers to zoom in for better detail.

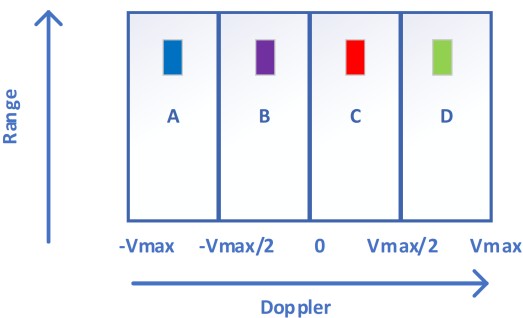

Figure S5: The illustration of Doppler Domain Multiplexing(DDM). This figure shows a toy example of how to use DDM to distinguish four transmitters.

less stringent IoU threshold, the recall of the RT-only variant experiences a significant increase. This result suggests that the RT-only variant's limitation mainly lies in its inferior angular localization ability. But by combining the strengths of camera and radar modalities, the model can leverage the advantages of both modalities, leading to a significant performance boost.

## D   Doppler domain multiplexing and coordinate system

As mentioned in [39], Doppler Domain Multiplexing(DDM) is used to distinguish received radar signals from different transmitters. The mechanism can be best illustrated in Range-Doppler (RD) spectrum format of data, as shown in Figure S5. Without DDM, the valid measurable speed interval is $[-V_{max}, V_{max}]$. Suppose we have four transmitters, Tx 1/2/3/4, and one object with speed in $[0, V_{max}/2]$. The echo signals from Tx 1/2/3/4 will respectively fall into sections C, D, A, B by using DDM, showing a periodical pattern. But now we can only measure speed in $[0, V_{max}/2]$. Similar

Table S5: BEV detection performances of different modality combination on KRadar Dataset test split with KITTI protocol of 40 recall positions. C, RA, RP respectively refer to the camera, range-azimuth map, and radar point cloud. The best result of each metric is in **bold**.

| Training set | Method | BEV AP(%)↑ | | | 3D AP(%)↑ | | |
|---|---|---|---|---|---|---|---|
| | | AP@0.3 | AP@0.5 | AP@0.7 | AP@0.3 | AP@0.5 | AP@0.7 |
| KRadar-20-train | EchoFusion(RP) | 55.74 | 43.94 | 20.56 | 53.40 | 29.21 | 2.90 |
| KRadar-20-train | EchoFusion(RA) | 54.45 | 42.28 | 17.97 | 51.75 | 24.65 | 3.85 |
| KRadar-20-train | EchoFusion(RP+C) | 66.46 | 54.17 | 26.39 | 58.62 | **34.67** | 3.96 |
| KRadar-20-train | EchoFusion(RA+C) | **68.70** | **55.90** | **29.65** | **66.68** | 34.33 | **5.34** |

phenomenon can be observed in the figure of Range Doppler data, i.e. the third image in Figure 2 in the paper.

We also carried out experiments on coordinate system of BEV space. Though the range dimension is under sphere system, we found that a polar system can achieve on par performance, as shown in Table S4. Thus we apply the polar system in the paper.

## E  More experimental results

Referring to Table S3, we have included the detection performance when our model consumes the ADC data or range-Doppler spectrum as input. It is worth noticing that we add an extra complex linear layer initialized with FFT coefficients for ADC data to replace the range FFT and the side-lobe suppression. In the case of the single modality version, the performance discrepancy between RT and these two modalities remains within the margin of error. When considering the fusion of image data, the performance of RD slightly lags behind that of RT, while that of ADC is almost the same as RT. These findings suggest that the integration of image data significantly mitigates the importance of conventional radar signal processing in accurate radar-based detection.

Besides, we offer ablation of different modality combinations on KRadar. The results are reported in Table S5. Here, we train all models with training split. The RA map shows considerable superiority over radar point cloud when fusing with the monocular image, which aligns with our research motivation and results on RADIal dataset.

