# OpenReview forum: "Echoes Beyond Points: Unleashing the Power of Raw Radar Data in Multi-modality Fusion"
_NeurIPS.cc/2023/Conference — NeurIPS 2023 poster_

### Official Review · Reviewer_mPFG · 2023-06-22

**Soundness:** 4 excellent
**Presentation:** 2 fair
**Contribution:** 4 excellent
**Rating:** 6
**Confidence:** 2

**Summary:**

This paper presents a method to perform image and raw radar fusion for object detection from a Mobile platform. The authors explain that the commonly used radar data processing pipelines discard too much data, resulting in very sparse point clouds. In contrast, the proposed method operates on raw data and performs end-to-end filtering within the model with intermodality cross-attention. They consider the RADIal radar dataset, for which they provide more precise and exhaustive annotations to better evaluate their method's added precision.

**Strengths:**

- The motivation and results are convincing, and the analysis interesting
- The model is natural and exploits the unique properties and structure of radar acquisitions
- Extensive ablation study and new annotations
- Not a lot of work on this interesting and promising modality exists



**Weaknesses:**

- The method section could be more precise and detailed. The CrossAttn operator is not defined with enough details (what are its arguments, Why are arguments 2 and 3 are always the same?), and neither are polar queries. There is a reference to [8] at the end of the method which helps a  little bit, but the paper should be readable on its own. As it stands, the cross-attention method is not reimplemetable.

- While submissions about sensors not typically treated at NeurIPS are welcome, the community is not necessarily familiar with the subtlety of radar data. As it stands, the submission overly relies on jargon and complex facts presented as self-evident. The figures and their caption could be more helpful. The authors should be more didactic.

- The writing could be significantly improved, as it contains many imprecise and clumsy formulations.

**Questions:**

This submission presents a well-motivated and designed method with promising results. However, the presentation makes it hard to follow the methodology, and the lack of pedagogy makes it hard to understand for non-radar experts. However, the weaknesses should be mostly fixable.

Q1) What are the arguments of CrossAttn? Why are arguments 2 and 3 are always the same?

Q2) What are polar queries? Are they different from object queries?

Q3) "The polar coordinate is aligned with radar data on the range dimension and aligned with image data on the azimuth dimension"
This sentence is hard to parse. What does this mean for a coordinate system to be aligned with a type of data?

Q4) Figure 1 is unclear and hard to parse. Which one is the proposed method? Whar exactly are the methods compared? What are the red circles supposed to highlight?

S1) Figure 2 could be great for explaining concretely the nature of the data presented. However, it is full of jargon and acronym and not clear at all. Maybe on a simpler example? Or even a toy example?

**Limitations:**

good

---

> ### Author Rebuttal · Authors · 2023-08-09
>
> Thank you for your thoughtful comments and advice. The responses to the main concerns are as follows:
>
> ## Q1: Clumsy formulations and jargons should be avoided
>
> Thanks for reminding! Our motivation involves some concepts of radar signal processing. We are trying our best to make them intuitive and vivid. We will adopt plainer expressions in the revision for easier understanding.
>
> ## Q2: What are the arguments of CrossAttn? Why are arguments 2 and 3 always the same?
>
> The Cross Attention is formulated as follows:
>
> $$
> \mathrm{CrossAttn}\left( q,K,V \right) =\mathrm{softmax} \left( \frac{\left( qW_Q \right) \left( KW_K \right) ^T}{\sqrt{d^*}} \right) \left( VW_V \right)
> $$
>
> $$
> where\,\,q\in \mathbb{R} ^{1\times d}, K\in \mathbb{R} ^{N\times d}, V\in \mathbb{R} ^{N\times d},W_Q\in \mathbb{R} ^{d\times d^*}, W_K\in \mathbb{R} ^{d\times d^*}, W_V\in \mathbb{R} ^{d\times d^*},
> $$
>
> here $N$ is the number of key vectors and value vectors. $d$ is the feature channel of input vectors. Matrix  $W_Q$, $W_K$, and $W_V$ are used to transform feature channel from $d$ to $d^*$.  The main idea of cross attention is first calculating similarity among query vectors $q$ and key vectors $K$. Then these similarities will act as weights to sum the value vectors $V$. The output vector is also of shape $\mathbb{R} ^{1\times d}$. Take Equation (3) in our paper as an example:
>
> $$
> \hat{q}\left( r_{bev},\phi_{bev} \right) =\mathrm{CrossAttn}\left( q\left( r_{bev},\phi _{bev} \right) , F_I\left( x_I \right) , F_I\left( x_I \right) \right).
> $$
>
> The second and the third arguments are the same means the key vectors $K\in \mathbb{R} ^{N\times d}$ and value vectors $V\in \mathbb{R} ^{N\times d}$  are the same.
>
> ## Q3: What are polar queries? Are they different from object queries?
>
> Polar Query and Object Query respectively depict the scene and candidate objects. To be specific, the polar query is of shape $\mathbb{R}^{R_l \times A_l \times d}$, which describes the whole scene covering the Field of View of radar sensor under BEV space, as mentioned in lines 159-161. On the other hand, each object query is described by a feature vector of $\mathbb{R}^{C}$ and a corresponding 3D position, which represents one candidate object. Thus, these two kinds of queries represent different entities.
>
> ## Q4: Understanding of “align with” in lines 44-46
>
> Sorry for the confusing description. Another reviewer also has a similar question. Please refer to “Understanding of “align with” in lines 44-46” part in General Reply.
>
> ## Q5: How to interpret Figure 1?
>
> Here the main purpose is to show the drawbacks of representing radar data as point cloud. Thus, we compare the detection results of different variants of our proposed method. The purple box denotes the result from the image-only variant of EchoFusion (row 1 in Table 5), the green box denotes the results from the EchoFusion variant that fuses imagery and range time data (row 6 in Table 5), while the blue box denotes result from image and radar point cloud fusion variant of EchoFusion (row 8 in Table 5). The red circles aim to highlight the false alarm radar points from conventional signal processing pipelines, which appear even if there is no object. And these false alarm points can lead to inaccurate estimation of the scale and position of targets.
>
> ## Q6: How to interpret Figure 2 and understand the mentioned jargon and acronym?
> Sorry for the confusing description. All the jargon and acronyms are explained in Section 3. They are the fundamental concepts in radar signal processing. In Figure 2 in the paper, **ADC** data means Analog-Digital-Converter (ADC) data, which is directly sampled from the received radar signal (mentioned in lines 115-121). **Range-Time** data means the vertical and horizontal axis respectively denotes range and radar signal periods. The generation of Range-Time data is mentioned in lines 124-127. **Range-Doppler** data means the vertical and horizontal axis respectively denotes range and speed. The generation of Range-Doppler data is mentioned in lines 128-130. **CFAR result** means the result of applying CFAR(Constant False Alarm Rate Detector, mentioned in lines 28-30) on the Range-Doppler data. **Radar point cloud** plots derived 3D radar points under Bird’s Eye View. The generation of the Radar point cloud is described in lines 137-139. **Range-Azimuth Map** means the vertical and horizontal axis respectively denotes range and azimuth. For clarity, we plot this map under polar coordinates. The generation of the Range-Azimuth Map is described in lines 139-142.
>
> In the Figure 2 caption in the paper, both the “Doppler Domain” and “Doppler dimension” in caption mean the horizontal axis, which indicates the speed of objects. We also mentioned Doppler domain multiplexing (DDM), which is used to distinguish received radar signals from different transmitters. As shown in Figure R2 in the pdf material, without DDM, the valid measurable speed interval is [ − *Vmax*, *Vmax*]. Suppose we have four transmitters, Tx 1/2/3/4, and one object with speed in [0, *Vmax*/2]. The echo signals from Tx 1/2/3/4 will respectively fall into sections C, D, A, B by using DDM, showing a periodical pattern. But now we can only measure speed in [0, *Vmax*/2]. Similar phenomenon can be observed in the figure of **Range Doppler data**, i.e. the third image in Figure 2 in the paper.

---

> > ### Comment · Reviewer_mPFG · 2023-08-14
> > **Follow up**
> >
> > I appreciate the comprehensive responses from the authors and the additional experiments, which provide valuable insights. The authors have effectively addressed many of the concerns that could be addressed as a rebuttal. However, a significant concern persists: the paper's readability. Some challenges arise from the writing, which could benefit from further refinement in termsof rigour, clarity, and didacticity. However, the primary obstacle is the inherent complexity of the RADAR modality.
> >
> > I encourage the authors to approach the presentation in a more pedagogical manner, without assuming the NeurIPS readership's familiarity with this specific modality. Given the considerable potential of RADAR and the contribution this paper brings to the table, I'm inclined towards recommending its acceptance, with the understanding that the authors will continue working on the paper's readability. Broadening the scope of accepted papers to include less modalities beyond text and image, like RADAR is the only way to incrrase the community's awareness, understanding and interest in these complex but powerful sensors.
> >
> > Wether the paper is accepted or resubmitted at another CV/ML conference, the authors should prioritize:
> > (i) Enhancing the writing quality to ensure it doesn't add to the subject's inherent complexity.
> > (ii) Articulating concepts as clearly as possible, prioritizing clarity without compromising precision. Deeper technical details can be relegated to the appendix for interested readers and radar experts.
> >
> > Regarding Q2, the uncommon nature of this approach warrants explicit mention to avoid any ambiguity.

---

> > > ### Author Response · Authors · 2023-08-14
> > >
> > > Thank you for your insightful comments and constructive suggestions. We do appreciate your willingness to recommend our paper for acceptance. We are going to add more pedagogical explanations of the RADAR modality and clarify related concepts as much as possible, so as to make the paper more accessible to a wider audience, especially for those without radar knowledge. Thank you again for your helpful comments.

---

### Official Review · Reviewer_w8tA · 2023-06-29

**Soundness:** 3 good
**Presentation:** 3 good
**Contribution:** 2 fair
**Rating:** 5
**Confidence:** 3

**Summary:**

This paper proposes a Polar-Aligned Attention module to guide the fusion of radar data and images. The module projects the image and radar features onto a polar coordinate system to calculate the similarity of the associated information. This effectively enhances the direct fusion of radar data and images. Additionally, the authors have restandardized the RADIal dataset, improving its accuracy and standardization.

**Strengths:**

The authors claim that this paper is the first work to directly fuse radar data with image data, achieving the fusion of relevant features through cleverly designed attention mechanism modules. Additionally, the authors' reannotation of the dataset has played a positive role in the development of this research direction.

**Weaknesses:**

Although this paper proposes a polar-coordinate-based attention mechanism module for the fusion of radar and image data, the contribution of such a structure is limited. Furthermore, due to the lack of dataset support for the research task of radar and image fusion in this study, the experiments were only conducted on an existing dataset with modifications. Therefore, I do not have enough confidence to affirm the sufficiency of the research motivation in this paper.

**Questions:**

1. How is the alignment of different modalities ensured when fusing the features of image and radar data?
2. Is it possible to visualize the alignment relationship between the two modalities in the polar-coordinate attention mechanism?

**Limitations:**

Due to the limitations of the dataset used in this study, it is not possible to extensively and sufficiently demonstrate the motivation behind the research in this paper.

---

> ### Author Rebuttal · Authors · 2023-08-10
>
> We are grateful for your valuable comments and constructive advice, which helps us a lot to make the paper better. The responses to the main concerns are as follows:
>
> ## Q1: Whether the novelty and contribution of this work is significant
>
> Indeed, our work is based on PolarFormer, but the fusion mechanism is newly designed because raw radar data has no explicit location information. Moreover, our efforts on the annotation is nonnegligible. Please refer to “The Contribution of This Paper” section in General Reply for a conclusion of our contribution.
>
> ## Q2: How is the alignment of different modalities ensured when fusing the features of image and radar data?
>
> The misalignment problem may come from numerous factors including calibration, sensor trigger, ego-motion, and other traffic participants’ motion. The calibration and trigger errors are determined by sensors and always ensured to be limited in an acceptable range. Thus the primary source of misalignments is ego-motion and other traffic participants’ motion. Since we didn't remove the distortion caused by motion, the misalignment mainly influence vehicles with high relative speed.
>
> We further examined misalignment caused by motion by measuring the average IoU of vehicles in the same (|yaw|<30 degrees) or opposite (|yaw|>150 degrees) directions with ego vehicle (yaw=0).  We take the IoU threshold of 0.5 and the score threshold of 0.01. The former situation includes 87 samples, with a recall of 70% and an average IoU of 0.799. The latter situation includes 1221 samples, with a recall of 94% and an average IoU of 0.867. It shows that vehicles with high relative speed lead to more severe misalignment, but our method still achieves relatively high IoU. Some bad cases are depicted in Figure R2 in the PDF of the general reply. Considering that the speed information is encoded in radar features, we believe that with more training data, the network is able to learn to better alleviate the misalignment by applying speed cues from radar.
>
> ## Q3: Is it possible to visualize the alignment relationship between the two modalities in the polar-coordinate attention mechanism?
> Thanks for your advice! We visualize the column-wise attention distribution in Figure R1 in the pdf file of general reply. As can be observed, the BEV queries learn to correctly attend to features of interested objects of specific range within corresponding image columns. But since we don't explicitly solve the azimuth information in range-time data and range-doppler spectrum, we are not able to get reasonable visualization of attention distribution for radar data.

---

> > ### Comment · Reviewer_w8tA · 2023-08-20
> >
> > I have referred to the author's response and the suggestions of other reviewers, but I still have concerns about the paper's contribution being sufficient for presentation at NeurIPS. Firstly, I acknowledge the author's re-annotation of the data as a contribution. However, the module proposed in the paper might lack the necessary theoretical depth. While I do recognize its effectiveness in visual tasks and the visualization clearly addressed my concerns, I am unsure of its novelty being suitable for a machine learning conference.
> >
> > In the end, I adjusted my score for the paper from 4 to 5, but I still retain my personal opinion.

---

### Official Review · Reviewer_eUpu · 2023-07-05

**Soundness:** 3 good
**Presentation:** 3 good
**Contribution:** 2 fair
**Rating:** 5
**Confidence:** 5

**Summary:**

This work presents a method for camera-radar detection and fusion. The authors propose an approach that follows:

Yanqin Jiang, Li Zhang, Zhenwei Miao, Xiatian Zhu, Jin Gao, Weiming Hu, and Yu-Gang Jiang. Polarformer: Multi-camera 3d object detection with polar transformers. AAAI, 2023.

and

Julien Rebut, Arthur Ouaknine, Waqas Malik, and Patrick Pérez. Raw high-definition radar for multi-task learning. In CVPR, 2022.

Specifically, the paper introduces a polar transformer method that cross-attends image and radar features across modalities according to view and sampling geometry. The technique relies on RAW radar measurements as input and, similar to recent works (above), is evaluated on an existing dataset. The approach is validated to be on par with recent methods.

**Strengths:**

The cross-modal fusion is well-motivated and has shown to be effective. Although a large body of work has proposed similar methods that also rely on RAW radar data, the design of the proposed method is sound and well-executed. Devising methods that operate on RAW data is an exciting opportunity for the field to embrace ideas from computational imaging in vision models.

**Weaknesses:**

The proposed method lacks novelty and experimental margins. Specifically, areas of concern are the following:

*Novelty and Technical Contribution: A large body of work has explored BEV-based fusion approaches and, recently, RAW-based detections. Specifically, this work directly follows the core ideas from:

Yanqin Jiang, Li Zhang, Zhenwei Miao, Xiatian Zhu, Jin Gao, Weiming Hu, and Yu-Gang Jiang. Polarformer: Multi-camera 3d object detection with polar transformers. AAAI, 2023.

and

Julien Rebut, Arthur Ouaknine, Waqas Malik, and Patrick Pérez. Raw high-definition radar for multi-task learning. In CVPR, 2022.

While the specific attention mechanism is new, it also extends the formulation in PolarFormer to the radar measurement formation model. As such, the technical contribution of the method is incremental in comparison to existing work in the field.

*Experimental Margins: As a results of the relatively incremental changes in architecture and training, the margins that the proposed method achieves are also also incremental. The method is on par with FFTRadNet on the original dataset, but improves after labeling. However, it is unclear if the baseline method has been retrained and all other baseline methods are also missing in the dataset. Moreover, without validating the relabeled dataset separately, it is unclear if the relabling biases the evaluation.

**Questions:**

Can the authors include examples of mislabeled objects or bias in the dataset?
What scenarios are corrected and why do they contribute significantly to the evaluation?

**Limitations:**

The authors discuss the limitations of their work and societal impacts of the method adequately.

---

> ### Author Rebuttal · Authors · 2023-08-10
>
> Thank you very much for your kind comments on our manuscript. The responses to the main concerns are as follows:
>
> ## Q1: Whether the novelty and contribution of this work is significant
>
> Indeed, our work is based on PolarFormer, but the fusion mechanism is newly designed because raw radar data has no explicit location information. Moreover, our efforts on the annotation is nonnegligible. Please refer to “The Contribution of This Paper” section in General Reply for a conclusion of our contribution.
>
> ## Q2: Uncertain about if the baseline method has been retrained and why all other baseline methods are missing in the newly labeled dataset
>
> To make a reasonable comparison with the baseline method, i.e. FFTRadNet, we added extra branches to predict the scale and orientation of targets (mentioned in lines 251-252), and this modified FFTRadNet is then retrained and tested on our relabeled dataset. The performance of this modified FFTRadNet is reported in Table 3 and discussed in Section 5.2 in the paper.
>
> Since the codes of other baseline methods are still not available (lines 252-253), we only compared their reported performance on original point-level annotation. The results are shown in Table 2 in the paper, which demonstrates the superiority of our method. And to facilitate further research, we are planning to open-source our code.
>
> ## Q3: What scenarios are corrected in new labels and how do the new labels influence the evaluation
>
> The original annotation and prediction of previous methods are center points of objects (mentioned in lines 75-77), and they assign a fixed scale and zero orientation when calculating Average Precision (AP). But vehicles of large angles and scales are normal in daily scenarios, as shown in Figure R3 in the PDF of the general reply. Our new annotations based on dense LiDAR points (mentioned in lines 75-77) correct this, helping a more comprehensive evaluation of existing methods. We also annotate other traffic participants such as pedestrians and cyclists. But due to their limited numbers (4 cyclists, 23 pedestrians in training split, and 17 cyclists, 0 pedestrians in test split), we don’t add more fine-grained class labels.

---

> > ### Comment · Reviewer_eUpu · 2023-08-20
> >
> > Thank you for conducting the additional experiments. My concerns are addressed and, hence, I have increased my rating. As the margins are still incremental, I have increased the ratings to a borderline accept.

---

### Official Review · Reviewer_K4kM · 2023-07-07

**Soundness:** 3 good
**Presentation:** 3 good
**Contribution:** 3 good
**Rating:** 5
**Confidence:** 4

**Summary:**

The paper proposes a method for fusing raw radar data with images in bird’s eye view (BEV) space. To this end, the method extract multilevel features from images and radar and uses polar-aligned attention mechanism with polar-BEV decoder for bounding box prediction. Evaluation is done on re-annotated RADIal dataset and shows better performance than other comparable methods.

**Strengths:**

The paper utilizes raw radar data which in general is a better approach than relying on post-processed radar data for object detection.


**Weaknesses:**

The BEV cross attention assumes synchronized radar and image which may not be robust at high speed driving and unsynchronized radar and camera setup.

**Questions:**

How robust is the proposed model to spatio-temporal misalignment?


**Limitations:**

The method assumes synchronized radar and camera.

---

> ### Author Rebuttal · Authors · 2023-08-10
>
> We sincerely thank you for the very professional comments. We are inspired and hope our discussion brings more insights. The responses to the main concerns are as follows:
>
> ## Q1: How robust is the proposed model to spatio-temporal misalignment?
>
> Both spatial and temporal misalignment can be alleviated by attention mechanism [1,3] and receptive field provided by convolution-based backbone [2]. As shown in Figure R1 in the pdf file of general reply, the BEV queries learn to correctly attend to features of interested objects of specific range within corresponding image columns. In RADIal, radar sample interval is 200ms, which is much larger than radar chirp period. Thus, it is hard to simulate temporal misalignment by concatenating radar chirps from nearby frames. So we only simulate spatial misalignment by adding noise to the extrinsic matrix. As shown in Table R2, our method shows strong robustness to the spatial alignment.
>
> Table R2. The detection performance under spatial and temporal misalignment. The spatial alignment is simulated by adding noise to the extrinsic matrix both in training and testing. Specifically, the rotation noises are sampled from a normal distribution with mean equals 0 and variance equals 2 (rotation noise are in degrees), and the translation noises are sampled from a normal distribution with mean equals 0 and variance equals 10 (translation noises are in centimeters, and the noise of each direction is independent).
>
> | Train | Test  | BEV AP/Overall | BEV AP/0-50m | BEV AP/50-100m | LET BEV AP/Overall | LET BEV AP/0-50m | LET BEV AP/50-100m |
> | --- | --- | --- | --- | --- | --- | --- | --- |
> | w/o noise | w/o noise | 84.25±0.33 | 86.67±1.17 | 90.48±1.04 | 88.54±0.52 | 92.10±1.15 | 94.45±0.17 |
> | w/o noise | w noise | 80.96 | 87.38 | 84.25 | 86.87 | 93.30 | 89.73 |
> | w noise | w/o noise | 84.65 | 88.27 | 90.84 | 88.29 | 91.99 | 95.68 |
> | w noise | w noise | 84.50 | 88.20 | 91.82 | 88.18 | 91.71 | 95.66 |
>
> > [1] Li Y, Yu A W, Meng T, et al. Deepfusion: Lidar-camera deep fusion for multi-modal 3d object detection, CVPR2022.
> >
> > [2] Liu Z, Tang H, Amini A, et al. Bevfusion: Multi-task multi-sensor fusion with unified bird’s-eye view representation, ICRA2023.
> >
> > [3] Li Z, Wang W, Li H, et al. Bevformer: Learning bird’s-eye-view representation from multi-camera images via spatiotemporal transformers, ECCV2022.

---

> > ### Comment · Reviewer_K4kM · 2023-08-21
> >
> > Thanks for the rebuttal and additional experiments. The attention mechanism seems to be robust to spatial misalignment to some extent and training with noise seems to be better in general.

---

### Official Review · Reviewer_jS6D · 2023-07-27

**Soundness:** 4 excellent
**Presentation:** 4 excellent
**Contribution:** 3 good
**Rating:** 8
**Confidence:** 3

**Summary:**

 The paper proposes a new method to fuse radar and imagery data for autonomous driving. Instead of using processed radar data, which discards most of the information in the original signal while attempting to filter out noise, this method leverages the raw data directly as input to the network. The method is based on attentional mechanisms cued to range in radar and azimuth in imagery, as these measurements are more accurate in those modalities. This technique and the use of raw data seem quite novel in this domain. The experiments are very thorough, on the only dataset that provides raw radar data. Full 3D bounding box annotations are added to this dataset, which is a useful contribution.

**Strengths:**

The introduction clearly lays out the motivation of using raw vs. processed radar data, and Fig. 1 provides a clear example of how this improves performance.

Lines 44-46 state “This technique is based on an observation that the polar coordinate is aligned with radar data on the range dimension and aligned with image data on the azimuth dimension.” It’s not clear with “aligned with” means here, but it seems you mean to say that radar data is more informative in the range dimension while imagery is more informative in the azimuth direction. It would be helpful to clarify this important point in the paper.

Adding vehicle bounding boxes to RADial is a significant contribution for those looking to study radar in autonomous driving. It would be useful to expand the annotations beyond a single “vehicle” class however, as with most of the other autonomous driving datasets.
The related work section is thorough and covers the main areas of claimed contributions.

The overall architecture, Fg. 3, nicely highlights the key contribution of the cross-attentional module in polar coordinates that emphasizes azimuthal accuracy in imagery and range accuracy in radar.

The approach in sec. 4 is clearly explained in text, supported by equations but without relying on those equations to encode the concepts, making the paper more readable and impactful.

The experiments are thorough, albeit on a single dataset as mentioned below. The primary comparative method [36], which operates on the processed radar data, performs considerably worse on the most important test, table 3, against the newly-created 3D bounding box annotations. The proposed method also does very well against all baselines using the standard point annotations on the original dataset.
The ablation studies are very thorough, examining performance on each sensor independently in order to show the value of fusion.


**Weaknesses:**

Experiments are performed on a single dataset, which is a minor limitation but unavoidable because it is the only one that provides raw radar data calibrated with imagery.

The focus of the paper is autonomous driving, which is a huge topic and is generally sufficient for papers at major AI conferences. Nevertheless it would be compelling to see how the method performs in other domains where radar and imagery could be fused, such as pedestrian detection and tracking, maritime vessel detection and so on. Datasets for these are hard to find, but perhaps the RADial dataset contains pedestrians that might have some motion signature in the radar data.


**Questions:**

As above.

**Limitations:**

Yes, no concerns.

---

> ### Author Rebuttal · Authors · 2023-08-10
>
> We really appreciate your affirmation of our work and constructive advice! The responses to the main concerns are as follows:
>
> ## Q1: Possibility of expanded annotation and tasks beyond vehicle detection
>
> In the original dataset there was only one class. In our refined annotation, we provide fine categories for objects such as car, truck, bus, cyclist and pedestrian. In the training dataset, there are 8227 instances, including 213 trucks or buses, 4 cyclists, and 23 pedestrians. In the test dataset, there are 1314 instances, including 63 trucks or buses, 17 cyclists, and 0 pedestrians. Since the number of these classes is quite limited and many of them stem from the same object, it is not sufficient to train a robust multi-class detector or make reasonable evaluation. Thus, we don’t offer fine-grained label annotations and only conduct experiments for one class. A larger and high-quality dataset is necessary for the research of raw radar data.
>
> ## Q2: Limitation of single dataset
>
> Recently we found another dataset contains raw radar data with camera and calibration. Although it has some limitations, we believe it would be a good complement to our work. Please refer to “More experiments on KRadar Dataset” part in General Reply.
>
> ## Q3: How to understand “align with” in the sentence “This technique is based on an observation that the polar coordinate is aligned with radar data on the range dimension and aligned with image data on the azimuth dimension.”
>
> Sorry for the confusing description. Another reviewer also has a similar question. Please refer to “Understanding of “align with” in lines 44-46” part in General Reply.

---

### Author Rebuttal · Authors · 2023-08-10

Dear all,

Thanks again for the very constructive comments and for spending your valuable time reading our responses! We are open to any further discussions during the author-reviewer discussion period (Aug 9-16). PDF material contains Figure R1-R4 used in personal replies.

## More Experiments on KRadar Dataset

Table R1. Detection performance on **KRadar-20** test split. AP_BEV and AP_3D are AP under Bird’s Eye View or 3D View. The ioU threshold of each metric is presented after @. Best results are in **bold**.

| Training Dataset | Vehicle AP/Conf@0.3 | AP_BEV@0.3 | AP_3D@0.3 | AP_BEV@0.5 | AP_3D@0.5 | AP_BEV@0.7 | AP_3D@0.7 |
| --- | --- | --- | --- | --- | --- | --- | --- |
|  KRadar | RTHN-Pretrained | 58.04 | 49.65 | 42.60 | 17.87 | 10.69 | 0.45 |
| KRadar-20-trainval | RTHN-Scratch | 61.38 | 53.05 | 46.47 | 17.98 | 10.47 | 3.03 |
| KRadar-20-trainval | EchoFusion-ra+img | 69.95(+8.63) | 68.35(+15.30) | 57.28(+10.81) | 43.87(+25.89) | 33.07(+22.60) | 14.00(+10.97) |
| KRadar-20-train | EchoFusion-rpcd | 55.74 | 53.40 | 43.94 | 29.21 | 20.56 | 2.90 |
| KRadar-20-train | EchoFusion-ra | 54.45 | 51.75 | 42.28 | 24.65 | 17.97 | 3.85 |
| KRadar-20-train | EchoFusion-rpcd+img | 66.46 | 58.62 | 54.17 | 34.67 | 26.39 | 3.96 |
| KRadar-20-train | EchoFusion-ra+img | 68.70 | 66.68 | 55.90 | 34.33 | 29.65 | 5.34 |

Through additional experiments on the recently published 4D Raddar dataset KRadar [1], we validate our motivation, and prove 3D prediction ability of EchoFusion with a high-resolution 4D radar and accurate 3D bounding boxes.

### About Dataset

We conduct more experiments on the recently published 4D Radar dataset [KRadar](https://github.com/kaist-avelab/K-Radar) [1]. However, we come across extremely limited speed when downloading the large-scale data(12T), which remains unsolved (refer to issues: [#2](https://github.com/kaist-avelab/K-Radar/issues/2),[#11](https://github.com/kaist-avelab/K-Radar/issues/11)). Thus we turn to 20 sequence sub-dataset stored on Google Drive (refer to issue [#9](https://github.com/kaist-avelab/K-Radar/issues/9)). This so-called **KRadar-20** dataset includes 6493 training samples and 6515 test samples. The 6 times larger test split than the RADIal dataset and 50% nighttime data make it challenging. The training samples are named as trainval split, it is further split into train split of 5190 samples and val split of 1303 samples for ablation.

### Result Analysis

KRadar dataset contains radar formats of range-azimuth map (RA map) and radar point cloud (radar pcd). Different from RADIal, the RA map of KRadar contains extra height dimension. We evaluate EchoFusion using these two types of data with official KITTI-style metrics of score threshold 0.3. The baseline is the official RTNH[1], which consumes radar pcd. To align with baseline, we take radius and azimuth range respectively as [0, 72] m and [-20, 20] degrees. The results are shown in Table R1. With trainval split and RA map and image fusion, our EchoFusion improves over 10 points at each metrics, except for BEV AP of IoU threshold 0.3. On ablation experiments using train split, the RA map shows considerable superiority over radar pcd when fusing with monocular image, which aligns with our research motivation.

> [1] Paek D H, Kong S H, Wijaya K T. K-Radar: 4D radar object detection for autonomous driving in various weather conditions, NIPS2022.
>

## The Contribution of This Paper

- We are the first to fuse raw radar data with images. Raw radar data has no explicit location information, which is significantly different from point cloud. Existing BEV methods cannot be directly applied in this task.
- We present the limitations of existing radar datasets and provide accurate 3D bounding box annotations for the raw radar dataset RADIal.
- We indeed demonstrate the advantage of using radar raw data instead of the converted radar point cloud in real benchmarks and achieve SOTA.

## Understanding of “align with” in lines 44-46

By using “align with” in this sentence “This technique is based on an observation that the polar coordinate is aligned with radar data on the range dimension and aligned with image data on the azimuth dimension.”, we mean that both of Range-Time data, Range-Doppler data, or Range-Azimuth Map have a dimension describing the range, which can be well depicted by the radial axis of polar coordinates. For the image, the horizontal dimension corresponds to azimuth, which can be well described using the azimuth axis of polar coordinates.

---

> ### Author Response · Authors · 2023-08-19
>
> Thanks again for your insightful comments and valuable time devoted to our paper. We are happy to answer any questions or concerns you may have during the author-reviewer discussion period (Aug 12-21).

---

### Decision · Program_Chairs · 2023-09-21

**Decision:**

Accept (poster)

**Comment:**

Already the initial reviews were rather positive, and after the rebuttal all five recommendations are on the positive side. The AC agrees with them. There was some criticism regarding the clarity of the paper, please make an effort to improve the presentation as promised during the discussion.